# Continuous Sign Language Recognition and Its Translation into Intonation-Colored Speech

**DOI:** 10.3390/s23146383

**Published:** 2023-07-13

**Authors:** Nurzada Amangeldy, Aru Ukenova, Gulmira Bekmanova, Bibigul Razakhova, Marek Milosz, Saule Kudubayeva

**Affiliations:** 1Faculty of Information Technologies, L.N. Gumilyov Eurasian National University, Astana 010000, Kazakhstan; 2Department of Computer Science, Lublin University of Technology, 36B Nadbystrzycka Str., 20-618 Lublin, Poland

**Keywords:** sign language recognition, natural language processing, intonational speech synthesis, long short-term memory, spatiotemporal features

## Abstract

This article is devoted to solving the problem of converting sign language into a consistent text with intonation markup for subsequent voice synthesis of sign phrases by speech with intonation. The paper proposes an improved method of continuous recognition of sign language, the results of which are transmitted to a natural language processor based on analyzers of morphology, syntax, and semantics of the Kazakh language, including morphological inflection and the construction of an intonation model of simple sentences. This approach has significant practical and social significance, as it can lead to the development of technologies that will help people with disabilities to communicate and improve their quality of life. As a result of the cross-validation of the model, we obtained an average test accuracy of 0.97 and an average val_accuracy of 0.90 for model evaluation. We also identified 20 sentence structures of the Kazakh language with their intonational model.

## 1. Introduction

This paper is a comprehensive study of several problems and includes research on the recognition of continuous sign language and natural language processing (NLP), using the example of the Kazakh language, including morphological, syntactic, and semantic analysis, morphological inflection, and the construction of an intonation model of simple sentences. 

The problem of continuous sign language recognition (CSLR), as well as the problem of generating intonation-colored speech, despite the amount of research conducted, remain to be resolved and have a wide range of practical applications.

Such an integrated approach is necessary since the solution of individual problems has only scientific value, and the solution of several problems creates the prerequisites for the authors to create a software product that can recognize sign language, translate it into text, and then, after NLP processing, create a consistent text in natural language and voice it using a synthesizer of intonationally-colored speech. In view of the large amount of research conducted, the synthesizer is not considered in this article, and only the intonational coloring of speech supplied as the input of the synthesizer is presented. By itself, sign language recognition in this study is significantly improved, and a new architectures for gesture recognition model is proposed.

CSLR and translating sign language into spoken language can enhance inclusion and accessibility for individuals who are deaf or hard of hearing. It allows them to communicate more effectively with people who do not understand sign language, enabling greater participation in various social, educational, and professional settings. Translating sign language into spoken language can bridge the communication gap between deaf and hearing individuals, making interactions more efficient. It reduces the need for intermediaries or interpreters, enabling direct and real-time communication. The objective of the study was to solve the problem of CSLR, processing, and modification into natural language sentences and translating it into intonation-colored speech. 

A review of CSLR literature [1,2,3,4,5,6,7,8,9,10,11,12,13,14,15,16,17] reveals a growing body of research focused on developing robust and efficient recognition systems. However, no work related to CSLR with intonation-colored speech has been carried out for the Kazakh language.

This study suggests a more effective method of CSLR, the output of which is sent to an NLP system based on analyzers of the morphology, syntax, and semantics of the Kazakh language, including morphological inflection and the construction of intonation models for simple sentences. The CSLR method is based on a 1024-unit LSTM model for the analysis of sequential gestures in the Kazakh language, combining the key points of body and hand gestures to obtain a complete picture of a person’s posture and hand movements. With formal rules on the basis of which a database (dictionary) of word forms of the Kazakh language with their complete morphological information was created, the system generates a given word form. Consequently, sound frequencies in each word class of the sentence were determined for sentence structure by observing the sound frequency of speech according to the sentence structure in real time as well as in recordings of the experiment that were made while the monitoring was in progress. 

In order to evaluate the model, we used cross-validation, which yielded an average test accuracy of 0.97 and an average val_accuracy of 0.90. Additionally, we created 20 Kazakh sentence structures together with their intonational models.

The paper contains an introduction, related works about CSLR and NLP with intonational methods, the methods of the current research, a discussion, and a conclusion of this study. Lastly, we will address the study limitations and future directions in this field. Moreover, the expected impact of the research is discussed.

## 2. Related Works

Gestures refer to nonverbal communication that makes it possible for people to express their thoughts, feelings, and emotions. Each gesture has a specific structure, which includes the shape of the hand, the position of the gesture in space and the way it is performed, the movement of the lips, and the emotions during the movement. The configuration of the hand refers to the specific position of the palm and the direction of the fingers. The location of a gesture in space is crucial to determining its meaning since it remains constant for each gesture. Gesture performance can be static, with a fixed hand shape, or dynamic, with a changing configuration of the hand in time and space. To recognize a static gesture, it is necessary to focus on determining the shape of the hand, while for a dynamic gesture, it is important to observe the movement of the hand.

By recognizing and analyzing gestures, it is possible to determine the corresponding words and phrases in traditional speech. For example, by analyzing the context in which gesture signs are used and the relationship between signs and their meanings, we can improve the accuracy of machine translation or make the continuous recognition of CSLR sign language possible.

CSLR is challenging due to the complexity and variability of sign language, as well as the need for accurate and reliable detection and recognition of hand, body, lip, and emotional movements. To overcome these problems, researchers are studying a number of approaches, including machine learning algorithms [5,6] and neural network models [7,8,9,10,11,12]. Recent advances in deep learning and neural networks have shown promising results in improving the accuracy and speed of CSLR systems.

To solve the CSLR problem, the authors of [3] proposed an innovative SignBERT system that solves these problems using high-quality video clips with an intelligent selection of key frames, (3 + 2 + 1)D ResNet for visual feature extraction, and a pre-trained BERT model for language modeling, including partially masked videos from isolated sign language datasets. The multimodal version of SignBERT includes hand images as an additional input, and an iterative learning strategy has been developed to fully utilize the system’s capabilities on the available data sets. Experimental results show the modern performance of CSLR. A new approach has been presented [8] for context-dependent continuous sign language recognition using a generative adversarial network architecture called sign language recognition generative adversarial network (SLRGAN). The proposed network architecture consists of a generator that recognizes sign language glosses by extracting spatial and temporal features from video sequences, as well as a discriminator that evaluates the quality of generator predictions by modeling textual information at the level of sentences and glosses. The authors of [9] introduced a new step-by-step CSLR system, whose ability to understand complex linguistic content was evaluated using a set of signed video sequences in Japanese sign language. The system consists of two main stages of processing: the initial automatic tense segmentation using the random forest binary classifier and the classification of words into segments using a convolution neural network. The main input data of the system are post-processed two-dimensional angular trajectories of the joints of the body, fingers, and face, extracted from the demonstrator’s video data using the OpenPose open-source library. The effectiveness of the system was evaluated with different types of data transformation and two different sets of word class labels. The best data transformations achieve high accuracy both for the segmentation stage (frame accuracy 0.92) and for the segment classification stage (accuracy 0.91).

A cross-modal approach [10] to learning is proposed, which uses textual information to recognize the state of sign language and its tense boundaries from poorly annotated video sequences. The method uses a multimodal transformer to simulate intra-class dependencies and improve the accuracy of network recognition. The proposed approach surpasses modern methods on three datasets (RWTH-Phoenix-Weather-2014 [13], RWTH-Phoenix-Weather-2014T [14], and CSL [15]) of CSLR. In this paper [11], we propose a continuous SLR system based on the merged ST-LSTM attention network. We call it Bi-ST-LSTM-A, and it bypasses the stages of sequence segmentation. The characteristics of the SL video are created by the convolutional neural network (CNN) two-stream model: one stream analyzes global motion information, and the other focuses on the local representation of gestures. ST-STM is used to merge spatiotemporal information, and then Bi-LSTM [12], based on attention, is used to measure the correlation between the video and the sentence. Finally, the conversion between the video and the sentence is established by the Bi-ST-STM-A network, and the recognition of the sentence is carried out using encoding and decoding operations.

There are also works on the recognition of gestures in various contexts, identification and authentication of people based on their physical characteristics, and determining the age of a person or his emotional state [16]. Additionally, a number of works described in [17] are devoted to human and robot interaction.

NLP processing methods are highly dependent on the language. The Kazakh language is reasonably standardized and falls within the Kipchak cluster [18], while Turkish is classified under the Oguz cluster of Turkic languages [19]. Similar to other Turkic languages, both Kazakh and Turkish possess an agglutinative structure. Like other Turkic languages, both Kazakh and Turkish exhibit an agglutinative nature, which involves the formation of words by attaching affixes (suffixes) to the root or stem of a word. This property allows for changes in meaning within the semantic category when new words are formed, while the structural category, represented by the ending, modifies the word composition without altering the meaning.

The paper [20] discusses the examination of morphological rules for a specific morphological analyzer, which utilizes a common algorithm for morphological analysis. The ontological model is also a tool for modeling the morphology of related languages [21]. The automation of word form generation in the Kazakh language was made possible by [22], where the authors formalized the morphological rules of agglutinative languages using a semantic neural network.

The methods for creating basic sentences in Kazakh were laid out in [23,24,25,26,27]. It was feasible to automate text parsing using these models [18,28] by using Chomsky’s context-free grammar (CFG) to formalize the syntax of uncomplicated sentences in the Kazakh language and create ontological models of these rules within the Protégé environment. This was done to facilitate syntax analysis, considering the semantic aspects of the constituent elements. The formal models developed can be used in various computer applications, such as grammar checking, semantic interpretation, dialogue comprehension, and machine translation.

The authors of [19] presented a rule-based method that utilizes a dictionary to perform sentiment analysis on texts written in the Kazakh language. The approach relies on morphological rules and an ontological model to achieve this analysis. Another study [27] uses morphological principles to perform sentiment analysis on Kazakh phrases. Additionally, the paper [28] suggests an examination of the semantics of basic Kazakh sentences through syntactic analysis. 

Another study [29] developed several syllable-structured Assamese word datasets from emotional speech recordings of both male and female speakers in the same age group in order to demonstrate the strong influence of emotions on intonational traits. A multi-style extractor is suggested by [30] to separate the embedding of style into two different layers. The final syllable level indicates intonation, while the sentence level depicts emotion. It uses relative attributes to describe intonation intensity at the syllable level for fine-grained intonation control.

In order to convey a speaker’s purpose, intonation becomes crucial. However, appropriate intonation is frequently not modeled by current end-to-end text-to-speech (TTS) systems. In order to solve this issue, [31] suggests an innovative, user-friendly technique for synthesizing speech in various intonations using predefined intonation templates. Speech samples are systematically grouped into intonation templates before TTS model training. 

Prosodic structure, which affects the way the listener perceives the speech flow, is just as important to the creation of satisfying synthetic sentence prosody as feelings or attitudes. The authors of [32,33] investigated the theoretical and technological causes of these issues and suggested a better feature engineering strategy for deep learning based on a different sentence intonation model, applied to French. The paper [34] illustrates the issue with automatic text segmentation into syntagms at the semantic and Belarusian’s level of punctuation. Long words are broken up and prosodic transcription is produced. For systems for Belarusian text-to-speech that make use of preset syntactic grammars in NooJ to produce better synthetic speech, its execution is crucial. Furthermore, [35] demonstrates that intonations represented by the generated pitch curves mirror those in the generated spectrograms.

The paper [36] offers a suggestion for a demonstration to enhance the synthesized speech’s prosody of mathematical markup language (MathML) based mathematical expressions, while [37] demonstrates that an end-to-end neural network with integrated second-order adaptable linear all-pole digital filters can produce intonation with a natural sound, provided that the proper stability conditions are applied. However, while intonational synthesis speech has come a long way in recent years, it still faces challenges in replicating the full range of natural intonation patterns and nuances of human speech.

A number of models [38,39,40,41,42,43,44] have been created for the speech recognition of the Kazakh language. The complexity with regard to Kazakh, its distinctive features, the scarcity of emotional speech datasets, and other factors make it difficult to develop a model for emotional speech detection in this language.

## 3. Methods

The research methodology and the results obtained can be applied to any language by using the language models of the selected language. Conditionally, the study can be divided into 3 tasks presented in Figure 1.

The semantic–syntactic frameworks of written and sign languages differ significantly, making it impossible to carry out an accurate translation of SL at the present moment despite the huge practical promise. The difficulty of continuous sign language recognition has so far remained unsolved. As a result, there are currently no models or methods for sign language translation systems that are fully automated. This study offers a novel way of selecting the most effective continuous sign language recognition strategy in order to achieve this. To do this, a multilayer perceptron of a newly selected, ideally designed architecture is trained using the stage-by-stage extracted properties of a person’s posture and hands. This method works by extracting the spatiotemporal characteristics of sign language using a multi-stage pipe. Prior to applying speech synthesis with intonation variation to improve communication, it is important to consider how the meaning of commonly used sign language expressions varies according to the right tense and case.

The problem of continuous sign language recognition has not yet been solved due to serious differences in the semantic and syntactic structures of written and sign languages, as a result of which it is not yet possible to perform an unambiguous translation of the language. Therefore, there are currently no fully automated models and methods for sign language translation systems. To this end, this paper presents a new approach to finding the best solution for continuous sign language recognition by extracting the spatiotemporal characteristics of sign words using a multi-stage pipe: First, step-by-step extracted signs of a person’s posture and hands are trained by a multilayer perceptron of an optimally selected new architecture. Next, there is an analysis of the meaning of the recognized phrases of the sign language and their change in accordance with the required tense and case before the synthesis of speech with intonation variation to improve the quality of communication.

### 3.1. Data

#### 3.1.1. Data for the CSLR Method

To collect visual–spatial characteristics, we applied a multi-stage approach to extracting gesture key points from video information using the MediaPipe Holistic conveyor. The pipe consists of several stages, each of which is designed to overcome the specific limitations of individual pose models and hand components.

The first step of the pipe is to evaluate a person’s posture using the Blaze pose detector and the subsequent reference model. This stage helps to identify key landmarks of the human body, such as the location of the head, shoulders, elbows, and knees.

Using the landmarks found in the first stage, the pipe defines two regions of interest (ROI) for each hand. These regions of interest are determined based on the location of a person’s hand in relation to his/her body. After determining the areas of interest, a reframing model is used to improve the regions of interest and ensure accurate capture of hand landmarks. Then, the full-resolution input frame is cropped to these regions of interest, and hand models specific to the hand gesture recognition task are used to evaluate the corresponding landmarks. This step makes it possible to accurately highlight key hand gestures, such as finger movements or hand shapes.

The combination of the pose and hand landmarks gives a total of 258 key gesture points. These points include:

Posture guidelines:A total of 33 x, y, and z values: these are the 3D coordinates (x, y, and z) of pose landmarks such as head, shoulders, elbows, and knees.A total of 33 visibility values: these are the visibility or confidence scores associated with each pose reference. They indicate how reliably the pose landmarks were detected.Total for position landmarks: 33 x, y, and z values + 33 visibility values = 66 values

Hand landmarks:4.A total of 22 x, y, and z values for both hands: These represent the 3D coordinates (x, y, and z) of the hand landmarks captured for each hand gesture. These landmarks can capture key hand movements and shapes, including finger movements.5.Total for hand landmarks: 22 x, y, and z values x 2 (for both hands) = 44 values

Combining the pose and hand landmarks, we get a total of 66 values for pose landmarks + 44 values for hand landmarks = 110 values. Thus, the final number of gesture key points is 258, broken down into 126 values for pose landmarks and 132 values for hand landmarks. This combination of landmarks allows one to get a complete picture of the posture and movements of a person’s hands.

The following individuals were involved in the process of collecting the dataset: a 9-year-old girl, a 12-year-old boy, a 40-year-old woman, and two 20-year-old girls. The dataset was designed to be diverse in terms of age and gender in order to create an efficient and gender-resilient model.

#### 3.1.2. Data for NLP Processing

The sentences generated by gesture recognition are delivered to the NLP processor and are composed of words in their base form. The root forms of words exhibit semantic characteristics including categories such as part of speech, animateness, and inanimateness for nouns, the production of comparative and superlative adjectives, the formation of cardinal and ordinal numerals, and the verb combination in complicated forms with auxiliary verbs. In the knowledge base, word modification and morphological analysis are performed using more than 30 semantic features in total. More than 3,200,000 dictionary entries are contained in the database that houses the dictionary. The system includes a dictionary in every language model. 

Our NLP processor contains a semantic knowledge base of formal principles of inflection, which is used as the foundation for creating a database (dictionary) of Kazakh word forms along with all of their associated morphological data. Semantic categories are also included in these formal rules. The following are snippets of formal inflection rules using a noun as an example. They take into account the law of synharmonism (vocalic harmony), which results in the addition of soft or hard ends depending on how soft or hard the word root is. As a result, there are 4500 formal rules for nouns, 96 for participles, 128 for adverbial modifiers, 13,000 for verbs, and a total of about 17,700 entries. The total of 3,200,000 word forms from the dictionary can be converted into around 56,000 root word forms using 17,700 formal criteria. This formal set of rules is used by the system to construct a certain word form. 

In order to recognize sign language, it is therefore desirable to employ either phrase recognition or recognition of simple sentences, hence a syntactic analyzer of Kazakh simple sentences is used. Like in English, the sequence of words in a sentence in Kazakh is very specific. A sentence often starts with a subject and concludes with a predicate, which makes syntactic analysis much easier. Twenty basic sentences can be found in the Kazakh language overall.

#### 3.1.3. Data for Processing of the Intonational Method

Twenty different types of basic sentences in the Kazakh language, including exclamatory and interrogative forms of the sentence, were given ten instances each in order to develop patterns of frequency modulation. The examples given were audio recorded, and the intonation characteristics and phonological norms of the language were taken into consideration [45,46,47,48]. Thus, 200 sentences were considered as subjects of the intonational variation model. The sound frequencies in each word class of the sentence were then identified for them by monitoring using the sound frequency analyzer website https://bellsofbliss.com/pages/sound-frequency-analyzer (accessed on 9 July 2023). By listening to recordings of the experiment that were created while it was being monitored, as well as by observing the sound frequency of speech in relation to sentence structure in real time. The experiment’s findings were used to create a sound frequency modulation of the sentence structure, which was then displayed as a graph. According to the sentence structures, we examine how the pitch varies at various points or intervals within a sound waveform to assess sound frequency variation over time. Through this study, we are able to comprehend the dynamics and patterns of pitch fluctuations inside a specific phrase structure sound.

### 3.2. CSLR Method in the Case of the Kazakh Language

In various studies, different sensors have been used to improve gesture recognition, but these systems have certain limitations. For example, some systems require invasive procedures, such as the use of electromyography (EMG) signals [49], which may limit their practical application in real-world conditions. Other systems use smartwatches [22], but they may not fully reflect the complexity and nuances of sign language gestures. The use of Leap Motion technology [50] in another system can create problems in accurately predicting finger positions. Meanwhile, using Kinect [51] in the context of recognizing sign language gestures provides significant opportunities for capturing and analyzing human hand and body movements. The convenience factor plays a decisive role in the adoption and widespread use of gesture recognition systems. If the integration of additional sensors and technologies leads to bulky or impractical setups, this may hinder the use and acceptance of these systems in real-world scenarios. Users may find it inconvenient to wear special gloves or devices, undergo invasive procedures, or deal with complex sensor settings.

Therefore, along with technological advancements, researchers and developers should also strive for convenient and unobtrusive solutions. This might include exploring alternative approaches that minimize dependence on additional sensors and utilize the capabilities of existing devices, such as smartphones or webcams. Computer-vision-based solutions using the built-in cameras of readily available devices may offer more practical and accessible gesture recognition options, including for continuous sign language recognition.

In this paper, a CSLR method is proposed using the example of the Kazakh language with the extraction of multimodal spatiotemporal features of dynamic gestures, by constructing an optimal neural network architecture (Figure 2). Various types of neural networks have been successfully applied in gesture recognition tasks; for example, CNNs [52,53,54] can automatically study the hierarchical features of raw images, which makes it possible for them to capture spatial information in gestures based on patterns obtained from training data. Deep neural network (DNN) is used for data-based training by changing the weights of connections between neurons in the training process [53]. Input data are sent to the first layer of the network, and each subsequent layer processes the output of the previous layer until the final result is obtained. The process of transmitting data forward over the network is called direct distribution. To process sequential data, such as tenses, long short-term memory (LSTM) is used in sign language recognition tasks [23,24,54], which provide feedback that makes it possible for them to retain the memory of past input data, and this is important for continuous recognition of gestures occurring over time.

LSTM is one of the feedforward neural networks. It is best suited for dealing with serial data. In the case of language gesture prediction, where each gesture is a sequence of frames, LSTM can easily model dependencies between successive frames and take into account the context, which allows the network to process and predict gesture sequences more efficiently. LSTM with 1024 units has more expressiveness and the ability to model complex data dependencies. However, it is worth considering that using a higher number of units may require more computing resources and time to train the model.

The obtained features were trained using a single LSTM layer with 1024 blocks, which accepts input data with a given input form and returns a sequence of outputs. LSTM can work with sequential or temporal data and has the ability to capture and use information about past events to make decisions in the present, which is important for continuous real-time gesture recognition. The data set contains 1495 gesture samples. Each gesture sample is represented as 60 frames, so the model is trained on a dataset containing 89,700 frames. Samples of gestures containing at least 60 frames can be fed to the model as input. The experiment started with a small number of units, for example, 64 or 128. Then, the number of units was gradually increased and the impact on the performance of the model was evaluated. Our model performed well on the training, test, and validation datasets with LSTM 1024.

Next, the dropout layer with a dropout coefficient of 0.5 discards (sets to zero) 50% of the inputs during training. This helps prevent overkill by forcing the model to learn more reliable representations of the input data. To distribute probabilities by classes, a dense layer with a softmax activation function is added (Figure 3).

For fast convergence and better performance, the Adam optimizer was used, and to measure the difference between the predicted probability distribution and the probability distribution of the output classes, the cross-entropy loss function was chosen.

In sign languages, signs consist of a combination of hand shapes, movements, and places, and they are not necessarily related to the grammatical structure of the spoken language. Thus, sign language signs are often shown in an infinitive form and are not conjugated or changed taking into account tense, person, or other grammatical features; therefore, 8 simple sentences were selected for the experiment (Table 1) and 23 roots of infinitives were extracted from the sentences under consideration (Table 2). 

To obtain an effective and gender-resistant model, each word from the sentence was recorded by 5 demonstrators (a 9-year-old girl, a 12-year-old boy, a 40-year-old woman, and two 20-year-old girls) 50 samples for a training sample, 10 samples for a test sample, and 5 samples for a validation sample.

To improve the performance of the model, when selecting data for the training sample, all records containing less than 60 shots were excluded, and, as a result, out of 50 originally recorded samples, the model was trained using the remaining samples, ranging from 28 to 50 (Figure 4).

Figure 4 illustrates the proportion of true positive results, which is 12 out of the 23 real positive outcomes. A high score of 0.97 indicates that our model is able to identify most samples. Despite the fact that demonstrators of different ages participated in the experiment, including children who could make mistakes when recording gestures, the model showed high results for the training sample, with a train_accuracy value of 0.99; for the test sample, with a test_accuracy value of 0.99; and for the validation sample, with a val_accuracy value of 0.92 (Figure 5). The frequently used leave-one-out [55] cross-validation method was also carried out to evaluate the model, where each sample is sequentially left as a test, and the rest of the data are used to train the model. As a result of cross-validation, the average test_accuracy is 0.97 and the average val_accuracy is 0.90.

As we can see from the error matrix, the largest error is 5 samples between the “grand_mother” and “rejoice” samples. Although these movements are visually identical when demonstrated in sign language, there aren’t any significant distinctions in the finger placements when done correctly, and we think young kids could make these mistakes. However, the visual representation of these motions in the “good” and “enjoy” examples for 4 samples of sign language are comparable, as is the meaning in natural languages.

Next, the CSLR system was implemented for the Kazakh sign language, which performs sign language recognition in real time using a pre-trained LSTM model. The system initializes the video capture object using the specified camera or a ready-made video file. Then, it uses the MediaPipe Holistic model to detect and track hand landmarks and pose landmarks in each shot of the video, extracts the key points of the hands, and forms a sequence of key points on a given number of shots. Next, the LSTM model is used to predict the gesture that will be made, based on the sequence of key points of the hand and the pose of the person. If the probability of the predicted gesture exceeds the specified threshold, then the gesture is added to the current offer. The current offer and the last recognized gesture and its probability are displayed in the demo window (Figure 6).

For each predicted word, a real-time probability value is output from the statement (Table 3). The system also allows the user to reset gesture recognition by pressing the ‘r’ key. In this case, the sequence of shots used as input data for the machine learning model is reset, as is the state of the machine learning model itself.

The model trained on a training sample excluding recordings with fewer frames achieved high accuracy rates on the test and validation samples. This indicates that the model successfully copes with the definition of gestures even with possible mistakes made when recording gestures by demonstrators of different ages.

The results of the system, using a pre-trained LSTM model, also show high accuracy of gesture recognition in real time, with the ability to display the current sentence consisting of words in the root form and probability values for each predicted word.

### 3.3. NLP Processing

Gesture recognition results are sentences that consist of words in the root form; they are transmitted to the NLP processor.

An NLP processor consists of the following blocks: morphological analyzer, syntax analyzer, and morphological corrector (word modifier).

The obvious complexity of processing natural language processes is caused by the difficulty of their formalization. The difficulty lies in the impossibility of word modification for any part of speech along a given trajectory without preliminary processing of the dictionary of root forms, since there is a dependence of word modification on its meaning, that is, on its semantic content. The semantic features of the root forms of words are such categories as part of speech, animateness, and inanimateness for nouns, the formation of comparative and superlative adjectives, the formation of cardinal and ordinal numerals, and verb combination in complex forms with auxiliary verbs such as “oтыр”, “тұр”, “жатыр”, “жүр”, etc. In total, there are more than 30 semantic features in the knowledge base, according to which word modification and morphological analysis are carried out. The volume of the dictionary exceeds 3,200,000 entries.

The Kazakh language belongs to the Turkic group of languages and is characterized by a large number of word forms for each word formed by adding suffixes and endings. Suffixes belong to the semantic category and when forming new words, they often change the parts of speech to which the root word or base belongs. For example, an indivisible root in the form of the verb “жаз—write” when adding the suffix “у” turns into a noun “жаз+у—writing” or into another verb “жаз+у—to write”, and adding another suffix “шы” to the latter turns it into a noun “жазу+шы—writer”. At the same time, adding an ending does not change the part of speech to which the base belongs (the indivisible root plus suffixes). For example, using the ending “лар”, we can get the plural of the noun “жазу+лар—writings, жазушы+лар—writers”.

The Kazakh language has a law of synharmonism (vocalic harmony) of sounds and syllables, which causes the addition of soft or hard endings depending on the softness or hardness of the base (indivisible root or root with a suffix), respectively [56].

Let us consider the example of the word “бала—child” (root form) and its two word forms “баламның—of my child”, “баламын—I am a child”. In the first case, two endings are attached to the base: the personal ending of the first person “-м” and the case ending “-ның”. In the second case, one personal ending of the first person “-мын” is added to the base. The search algorithm should provide for any possible number of attached endings and accumulation of morphological information.

In the case of recognizing sentences that consist of words in the root form, the task of morphological analysis is simplified. Nevertheless, in this task, we use a standard morphological analysis algorithm.

#### 3.3.1. Morphological Analysis Algorithm

Morphological analysis algorithm: The word is being read;The dictionary of root forms opens and the search for the read word is performed in it;If the word is found, then go to step 8, otherwise step 4;The word in the loop is read character by character, starting from the last character, and we look for what we get in the dictionary of endings;If the ending is found, then we look for the remainder in the dictionary of root forms;Memorize the morphological information of the word;If such a word is not found, then go to step 4, otherwise to step 8;The end.

For example, for the sentence “БАЛАЛАР ЖАРЫҚ БӨЛМЕ ОТЫРУ”, the translation and writing in natural language are presented in Table 4.


The morphological analysis algorithm will return the following results:

Балалар (CHILDREN)—noun, animate, plural, nominative case;

ЖАРЫҚ (LIGHT)—adjective; 

БӨЛМЕ (ROOM)—noun, animate, singular, nominative case;

ОТЫРу—a verb in its root form.

#### 3.3.2. Syntactic Analyzer 

In this paper, a syntactic analyzer of simple sentences in the Kazakh language is used, since for sign language recognition, it is preferable to use either phrase recognition or recognition of simple sentences. In the Kazakh language, as in English, the order of words in a sentence is strictly defined. As a rule, a sentence begins with a subject and ends with a predicate, which greatly simplifies the task of syntactic analysis. The total number of simple sentences in the Kazakh language is 20. Below are all 20 given in a formal bracket entry. 

To understand the ways of formation of the Kazakh language, we present morphological and syntactic names and symbols in Table 5. These symbols were taken in accordance with the names and symbols used by the Kazakh language in the UniTurk metalanguage, which describes the morphology, syntax, and semantics of Turkic languages [45].

We show the ways of forming simple sentences (Table 6) in the Kazakh language according to the semantic model in brackets.

Since each sentence member has a strictly defined place and 1 or more variants of a part of speech, based on the described formal rules, the syntactic analyzer concludes what kind of sentence it has analyzed and determines the members of the sentence.

In our example, the simplified syntactic analyzer algorithm will return the following results:

Балалар (CHILDREN)—the subject

ЖАРЫҚ (LIGHT)—the attributive

БӨЛМЕ (ROOM)—the object

ОТЫРу (SIT)—the predicate

Which corresponds to the simple sentence number 7:

7) According to the subject, the attributive, the object, and the predicate:

SS(Q(Q7(Sub Att Obj Pre)) M(M7(MSub (N Pron Adj Num) MAtt(Adv Pron Num Adv) MObj (N Adj Num) MPre(V)))).

This algorithm takes into account the fact that, for example, the object cannot be in the nominative case, since sign language does not take cases into account. Moreover, it gives us the result that the object has been found, the case of which should be changed in this example to the dative case.

However, in order to bring this sentence into the form in which it is used in natural language, we need to modify the object, namely БӨЛМЕ in БӨЛМЕде by adding the ending -де. This is done with the help of a morphological corrector. 

#### 3.3.3. Morphological Corrector (Word Modifier)

Any language model of the system includes a dictionary. In our NLP processor, there is a semantic knowledge base of formal rules of inflection, on the basis of which a database (dictionary) of word forms of the Kazakh language with their complete morphological information is generated. These formal rules also contain semantic categories. The following are fragments of the formal rules of inflection using the example of a noun, taking into account the law of synharmonism (vocalic harmony), which causes the addition of soft or hard endings depending on the softness or hardness of the word base.

The given example shows a fragment of the rules, where “зе” means зат есім (the noun), “жа” means жанды (animate), “01” ends with the hard vowels а, o, ұ, “))” indicates there are endings of nouns between the closing brackets, and after “!” there is morphological information [28]. All word forms are for one animate noun (Table 7).


The result is the following statistics:
− The number of formal rules for the noun is 4500;− The number of formal rules for the participle is 96;− The number of formal rules for the adverbial modifier is 128;− The number of formal rules for the verb is 13,000;− The total number of formal rules is about 17,700 entries.

17,700 formal rules allow the generation of 3,200,000 word forms from the dictionary into about 56,000 root word forms. 

Based on these formal rules, the system generates a given word form. In our case, the sentence “БАЛАЛАР ЖАРЫҚ БӨЛМЕ ОТЫРУ” is changed to “БАЛАЛАР ЖАРЫҚ БӨЛМЕде ОТЫР”, so it fully corresponds to the grammar of the natural language and can be voiced by a synthesizer.

### 3.4. Intonation Variation Model

This article discusses the methodology of the synthesis of intonation-colored Kazakh speech. Intonation synthesis has not been carried out for the Kazakh language and the proposed methodology is the first such attempt. To solve this problem, 20 simple sentences of the Kazakh language were analyzed and changes in intonation during their pronunciation were experimentally measured. The main problem to be solved is avoiding the monotony of the text read by the synthesizer, as well as the likelihood of intonation.

Sentences play an important role in emotional speech recognition by providing a larger context for interpreting emotional cues in speech. Machine learning models can interpret the speaker’s emotional state more precisely by analyzing the structure and substance of sentences in order to comprehend the speaker’s intended meaning. Due to this, the structural model of the sentence which is specific to the Kazakh language will be used to create a model of emotional speech recognition.

Sound frequency is also a key factor in the recognition of emotional speech because it includes important information about the acoustic features of speech that can communicate emotional cues. According to the Kazakh linguist Z.M. Bazarbayeva’s findings [46,47,48], intonation can be used to shape oral speech and bring out the communicative-pragmatic and emotionally-expressive aspects of expression. Using acoustic factors such as fundamental tone frequency, tempo, and intensity, speech sounds and intonation perform a communicative function (timbre). Since certain feelings frequently correspond with particular patterns of frequency modulation, the frequency content of speech in particular can shed light on the speaker’s emotional state.

The changes in pitch or sound frequency that take place within a sentence are referred to as the sound frequency variation method of sentence structure [47]. This paradigm is essential for expressing intonation, adding nuance, and conveying meaning in speech. A key element of intonation, which relates to the rise and fall of pitch patterns in speech, is sound frequency fluctuation. It aids in expressing the rhythmic and melodic contours of phrases. Different pitch patterns can signify different sentence kinds (declarative, interrogative, exclamatory), as well as express various pragmatic meanings [57]. Additionally, the emphasis on particular words or phrases within a sentence can be achieved through sound frequency change [58]. Speakers can emphasize contrastive aspects or attract attention to essential information by changing the pitch or sound frequency of specific words. These differences in pitch produce pitch accents, which support the sentence’s overall prosody and meaning [59].

Moreover, variations in sound frequency are essential for expressing emotion in speech. Pitch and sound frequency variations can express emotions such as elation, happiness, rage, grief, or unease [46,60]. The emotional undertone of the statement is influenced by the way the pitch increases, decreases, or remains constant.

In addition, variations in sound frequency can assist in clarifying sentence meaning [45]. For instance, in some languages, a word’s meaning or grammatical function can be altered by changing the pitch contour of that word [61]. Speakers are able to distinguish between statements, questions, and other sentence kinds by employing various sound frequency differences.

The hierarchical arrangement of phrases inside a sentence is influenced by sound frequency fluctuation. It offers indications for grouping and hierarchy and helps draw lines between words, phrases, and clauses [62]. Speakers build a prosodic structure that facilitates understanding and meaning interpretation by using pitch fluctuations [63].

In order to improve the chance of natural intonation and prevent the monotony of text read by a synthesizer, the sound frequency variation approach was developed [64]. Making speech sound expressive and natural rather than robotic or repetitive is one of the difficulties of speech synthesis. It is possible to get more realistic and varied intonation patterns in synthesized speech by using sound frequency fluctuation [65].

If we only exhibit sound frequency in relation to phrase form, we can obtain a number of outcomes [45,66]. To begin with, prosodic patterns can be discovered by examining the sound frequency fluctuations within phrase forms. By observing the frequency patterns, we may spot sections of elevated pitch, which denote the areas of the phrase that are highlighted or contain the most significant information [67]. Additionally, disfluencies and speech mistakes within phrase structures can be found using frequency analysis [46,48]. The frequency patterns may be disturbed or deviate when people stumble or make mistakes while speaking [46,47]. To better understand speech production and communication issues, we can use these frequency irregularities to investigate the frequency and type of disfluencies. Finally, sound frequency analysis can make specific speech traits intelligible. Depending on their vocal habits and qualities, different speakers may display distinctive pitch patterns or frequency contours [68]. We can investigate speaker-specific features and possibly use them for tasks such as speaker identification or speech recognition by researching these individual variances. The intonational variation model of sentence structure and sound frequency are closely connected. Sound frequency, as used in relation to speech, is the rate at which the vocal cords vibrate and generate sound waves. For spoken language to express intonation and meaning, changes in sound frequency, or pitch, are essential.

According to the intonational variation model, changes in pitch and melody within a sentence are taken into account, and these variations are mirrored in the sound frequencies generated [69]. Therefore, the intonational variation model has a relationship with sound frequency. In order to create patterns of frequency modulation, 10 examples for each of the 20 types of simple sentences in the Kazakh language, including exclamatory and interrogative forms of the sentence, were given. It should be noted that there are basically four types of sentences in the Kazakh language. However, an imperative sentence can be classified as a declarative or an exclamatory sentence depending on the purpose of its utterance. Therefore, three types of sentences are used in this research work. First, taking into account the intonation features and phonologic rules of the language [28,46,47,48], the given examples were recorded on audio, and then the sound frequencies in each word class of the sentence were determined for them through monitoring with the sound frequency analyzer website https://bellsofbliss.com (accessed on 9 July 2023). The sound frequency of speech according to the sentence structure was observed in real time as well as on recordings of the experiment which were made while monitoring. Using the results of the experiment, a sound frequency modulation of the sentence structure was made and presented in the form of a graph. We analyze sound frequency variation over time by examining how the pitch changes across different moments or intervals within a sound waveform according to the sentence structures. This analysis allows us to understand the dynamics and patterns of pitch fluctuations within a given sound of the sentence structure.

Figure 7 shows the sounding frequencies of sentence structures 1 and 2. A model of sentence structure 1 includes word classes that can be the primary sentence part; according to the research methodology, 10 examples of the mentioned sentence were given, and the pronunciation frequencies were monitored. In the form of a reporting sentence, the mentioned part of the sentence has a frequency of 510–850 Hz, in the form of an interrogative sentence it is raised to 1200 Hz, and in the form of an exclamatory sentence, it is sounded at a frequency of 2200–3100 Hz. However, in sentence structure 2, the predicate came together with the part of the sentence, and the highest value changed to 1750 Hz. Generally, it should be noted that, as a result of the experiment, the declarative, interrogative, and exclamatory types of sentences were distinguished by the frequency of the predicate, depending on the intonation and the purpose of the utterance. For example, in sentence structure 2, the sound of the predicate was in the range of 800–1000 Hz in the form of a reporting sentence, and 750 Hz in the interrogative form. However, in the form of an exclamatory sentence, it rose to 1100 Hz.

The results of sentence structures 3 and 4 are shown in Figure 8. In the cases where there are object or adverbial modifier members between the subject and predicate members, the subject and predicate members showed similar values in sentence structures 3 and 4. For example, in sentence structure 3, the subject range was between 515–585 Hz, and in sentence structure 4, the result was between 700–800 Hz. The predicate sounded at 850–1000 Hz in the form of a declarative sentence, 720–750 Hz in the form of an interrogative sentence, and 850 Hz in the form of an exclamatory sentence. In addition, if the object is located between the subject and predicate, it was performed in the range of 595–745 Hz, and the adverbial modifier showed a result in the range of 1500–2000 Hz.

The results of an object and an adverbial modifier placed between the subject and the predicate are graphed in Figure 9. In sentence structure 5, the subject at the beginning of the sentence and in front of the object was sounded at the frequency of 750–900 Hz, while the object showed results in the frequency range of 2000–3000 Hz. While the following adverbial modifier sounds around 1000–1800 Hz, the predicate showed sound frequencies of 850–1000 Hz, but it decreased to 750 Hz in the interrogative form and sounded up to 850 Hz in the form of an exclamatory sentence. The structure of sentence 6 is similar to the structure of sentence 5, but the positions of the adverbial modifier and the object members have changed. According to the results of the experiment, the sounding frequencies of the subject and the predicate in sentence structure 6 are similar to the values in sentence structure 5. However, the object was sounded in the range of 950–1500 Hz, and the adverbial modifier was sounded in the range of 1500–2000 Hz. It should be noted that the predicate’s reported form provided a value as low as 835 Hz.

In sentence structures 7 and 8 (Figure 10) there are constituent and auxiliary members of the sentence such as the attributive and the object. In sentence structure 7, the subject is in the range of 650–800 Hz, the voiced attributive is performed at a frequency of 960–1200 Hz, and the object has changed to 750–870 Hz. A predicate coming at the end of a sentence was reported at 850–1000 Hz. An object at the beginning of a sentence varied between 720 and 860 Hz and affected an attributive following it to increase to 950 Hz. The detectable subject sounded between 730 and 860 Hz, and the predicate changed to a frequency between 700 and 800 Hz. The predicate in the interrogative sentence dropped to 750 Hz, and in the exclamatory sentence it sounded between 815 and 950 Hz, and these indicators were similar in sentence structures 7 and 8.

If the object at the beginning of sentence structure 9 was sounded with a frequency of 860–935 Hz (Figure 11), the adverbial modifier that determined its characteristic or causal purpose was at a frequency of 500–650 Hz. Moreover, if the subject made up 820–970 Hz, its action was sounded by decreasing to 350 Hz. However, it increased to 598 Hz in the form of an interrogative sentence and changed to 650 Hz in an exclamatory sentence. The object of the sentence structure started with a frequency of 650–750 Hz and contributed to the performance of the subject in the range of 850–950 Hz. A circumstance sounded with a frequency of 1400–2000 Hz caused the predicate to change between 850 and 500 Hz. The mentioned predicate changed to 610 Hz in the interrogative sentence and 750 Hz in the exclamatory sentence.

Sentence structure 11 expressed the periodic location of the action from the adverbial modifier at the frequency of 900–1050 Hz and contributed to the implementation of the subject at the frequency of 850–1000 Hz (Figure 12). An object voiced at 750–875 Hz provided a change in the predicate between 900–750 Hz. This indicator led to a decrease to 517 Hz in the interrogative sentence and to 456 Hz in the exclamatory sentence. If the attributive member of sentence structure 12 begins with a frequency of 700–880 Hz, one of the constituent members of the sentence is the subject that expands with a frequency of 600–800 Hz. The 850–970 Hz object complements it and clarifies the function of the 720–800 Hz predicate. In interrogative and exclamatory sentence types, this value did not change significantly, but changed to 744 Hz and 780 Hz, respectively.

In a sentence structure consisting of 4 members, the attributive with a frequency change of 2000–2700 Hz gave tone to the subject with a frequency of 800–900 Hz (Figure 13). This idea led the predicate to sound in the range of 850–965 Hz with a frequency of 750–850 Hz. It can also be seen that the predicate has changed to 717 Hz in the interrogative sentence and 626 Hz in the exclamatory sentence. Sentence structure 14 began with a subject frequency of 500–620 Hz, and the adverbial modifier with a frequency of 1300–1600 Hz was followed by to an attributive (750–900 Hz). That is, the attributive that came before the object with a frequency of 700–810 Hz was sounded at a lower frequency than in the previous sentence. The adverbial modifier, which comes between the subject and the attributive, is performed in the frequency range of 1300–1600 Hz. It can be seen that the remaining members of the sentence correspond to the values in the previous sentence structures.

Figure 14 shows the results of sentence structures 15 and 16. It can be seen that the sentence starting with the subject with a frequency of 400–550 Hz has a high-frequency object (950–1160 Hz). Further, this indicator contributed to the sounding of the predicate in the range of 550–700 Hz with the frequency of 750–850 Hz of the adverbial modifier. However, in the interrogative sentence, the frequency increased to 760 Hz, and in the exclamatory sentence to 815 Hz. Meanwhile, in sentence structure 16, the adverbial modifier came before the other members of the sentence and was sounded at a high frequency of 2400–3000 Hz; if it changed the subject that came after it to 1000–1200 Hz, the attributive continued the sentence with the frequency of 750–860 Hz, followed by the object at 850–950 Hz. The predicate which ends the sentence is sounded at a frequency of 750–850 Hz, and the frequency of sounding is reduced in the interrogative (780 Hz) and exclamatory sentences (720 Hz).

Sentence structure 17 begins with an adverbial modifier with a frequency of 800–920 Hz (Figure 15). Further, the object continues with a frequency of 950–1100 Hz, while the attributive with a frequency of 750–920 Hz tones the subject (650–715 Hz). The predicate ends the sentence with a frequency of 700–745 Hz; this indicator increased to 817 Hz in the interrogative sentence, while in the exclamatory sentence, it sounded around 713 Hz. There is a difference in sentence structure 18 compared to the previous sentence structure. That is, the arrival of the attributive and the subject at the beginning of the sentence affected the sounding of the subject in the range of 820–900 Hz. The predicate was at a frequency between 750–800 Hz, and if it changed to 890 Hz in the interrogative sentence, it decreased to 750 Hz in the exclamatory sentence.

In Figure 16, the replacement of the members of the sentence showed different results. For instance, in sentence structure 19, a high frequency was observed in the attributive (850–950 Hz) and object (820–920 Hz), while in sentence structure 20, the attributive changed to a frequency of 1000–1220 Hz, and the frequency between 850–920 Hz was characteristic of the subject. In this sentence, the predicate sounded between 950–820 Hz, changing to 760 Hz in interrogative sentences, and 650 Hz in exclamatory sentences.

Thus, the order of sounding of the members of the sentence, which are basically word classes, was at different frequency intervals. We can verify the results of this study by applying them to the experiment in Section 3.2. Figure 17 shows the 2 sentence structures of the mentioned experiment. These sentences correspond to sentence structures 2 and 3 and the sounding frequencies of the sentences are around the indicators shown in Figure 7 and Figure 8.

Let us consider the 2 sentence examples from the CSLR experiment, corresponding to sentence structure 5 (Figure 9). As can be seen from Figure 18, the sounding frequencies of the parts of the sentence do not deviate from the results of sentence structure 5. Therefore, the correctness of the research results can be observed again. 

Figure 19 shows other sentences of the CSLR experiment corresponding to sentence structure 7. The subject sounded in the range of 650–720 Hz, while the predicate sounded in the range of 850–1000 Hz, and these results are consistent with the results of the study in Figure 10. The attributive parts were performed in the range of 960–1110 Hz, while the object sounded with 750–820 Hz.

Figure 20 shows an example of a sentence corresponding to sentence structure 10 from the experiment in Section 3.2. The object that came at the beginning of the sentence was sounded with a frequency of 650–750 Hz and affected the performance of the subject at 900–950 Hz; the adverbial modifier sounded around 1750–2000 Hz and contributed to the sound of the predicate by changing from 850 Hz to 750 Hz. These values correspond to the indicators in Figure 11b.

In this section, the results of the CSLR experiment are shown in intonation-colored speech by using the research results related to the intonational model of Kazakh sentences, and it can be seen that the integrity of the mentioned research work has been proven.

## 4. Discussion

The proposed method can be used to develop effective automated continuous sign language translation systems that can facilitate communication and interaction of deaf and hard-of-hearing people with society by synthesizing speech with intonation variation to improve the quality of communication.

The test results show that the system successfully predicts simple sentences as a sequence of words in the root form. 

It is important to note that the success of the approach depends on various factors, such as the quality and quantity of data, and the age and gender of the demonstrators used to train the model. Therefore, each gesture was recorded 40 times, by 5 demonstrators (a 9-year-old girl, a 12-year-old boy, two 20-year-old girls, and a 40-year-old woman), resulting in 23 gesture words of 50 copies, each copy having 60 shots.

According to the results of the research, the sound frequency of a sentence changed depending on the location and number of the members of the sentence. Additionally, it is well known that the predicate’s frequency varies based on whether the sentence is of the informative, interrogative, or exclamatory type and its intonation. For instance, sentence structure 1 consists of only one part of the sentence, and in the three types of the sentence, one can observe large frequency changes. The mentioned sentence construction was sounded at a frequency of 510–850 Hz in the declarative sentence, 850–1200 Hz in the interrogative form, and 2200–3100 Hz in the exclamatory sentence. 

It is evident from sentence structure 2, which is made up of the sentence’s constituent members, that the subject is used frequently in speech (up to 2000 Hz). Additionally, auxiliary members of sentences tended to occur more frequently. At least one of the unsettling members described above was audible at a higher frequency.

If we focus on the change depending on the number of members in the sentence, in the structure consisting of four members, the object came after the subject and was sounded at a high frequency, in the range of 2000–3000 Hz. Meanwhile, if the adverbial modifier is sounded with a frequency of 1000–2000 Hz, the attributive comes at the beginning of the sentence and is spoken between 2000–2700 Hz. A sentence consisting of five members had an adverbial modifier that was sounded with the highest frequency (Figure 16b).

## 5. Conclusions

A particular feature of this work is an integrated approach to solving the problem of recognizing continuous sign language and translating it into a natural language using morphological, syntactic, and semantic analysis, as well as building an intonation model of simple sentences for subsequent synthesis. Such an integrated approach makes it possible to perform the important task of creating an intonation speech synthesizer based on a recognized sign language. This is the practical and social significance of this study. The research methodology and the results obtained can be applied to any natural language in the future.

The scientific novelty of the work is a new method of continuous sign language recognition, which combines several modalities, including hand movement, hand configuration, and body and head movements, which increases the accuracy of sign language recognition. By integrating information from various modalities, the proposed LSTM (1024) architecture can better understand the nuances of sign language and classify them with a high accuracy of 0.95. After integrating the proposed model into the CSLR system of the Kazakh language, simple sentences were obtained that consist of words in their root form with a high average probability value of 0.92.

Another innovation of the work is the integration of a sign language recognizer with an NLP processor and the translation of recognized sign language sentences consisting of words in the initial forms of the word into fully consistent sentences.

Moreover, in this paper, for the first time, a study of the intonation of the Kazakh language is presented depending on the change in the frequency of the main tone and sentence member for all types of simple sentences of the Kazakh language, which will allow synthesizing intonation-colored speech in the researchers’ future works.

Limitations: One of the main limitations when recording gestures is the quality of the recording itself. Insufficient camera resolution below 480 × 640 or incorrect positioning of the recording device may result in the loss of details and information about gestures, making them difficult to recognize correctly. Therefore, in our system for recording gestures, there is a draw_people functionality that allows us to get approximately equal distance from the camera and the angle of the dataset demonstrator [2]. Moreover, low lighting, the presence of interference, or noise in the recording can also have a negative impact on the accuracy of gesture recognition. In addition, our model has a limitation, according to which the sample must contain at least 60 frames. If the sample contains fewer frames, the system automatically filters it. The number of frames can be changed depending on the specific task and system requirements.

Expected impact: The expected effects are manifold. First, at the scientific level, the proposed methods can be adapted for most sign languages, can be improved, or serve as a point of comparison and improvement for existing methods. The indisputable advantage of this work is the complexity of the study of sign language and its projection on natural language, emotionality, and the intonation of natural language, and the fact that this study is being carried out as part of a large funded project, that is, the ability to connect it with another task of the project in the form of speech synthesis and complete this study as a final product with the potential for commercialization.

The impact of the project at the social level lies in that in view of the fact that a large number of people with disabilities associated with the peculiarities of their hearing and speech need various applications that improve their quality of life and improve the possibilities of barrier-free communication, the results of this study may have a positive social impact, both nationally and internationally, as the results can be disseminated in most countries of the Central Asian region. At the national level, the project can lead to a social impact, such as the inclusion of people with speech impairments, improved communication in healthcare, education, and daily life, as well as the creation of support technologies that promote independence and improve the quality of life.

The project has the potential to generate positive economic effects, including the emergence of new markets and business opportunities, the growth of the healthcare industry through improved quality of service, and the emergence of new educational services.

Future works: The proposed method will serve as a prerequisite for sign language translation systems for media products, displaying television content, and communicating with deaf and dumb passengers in transport systems (airports, public transport, etc.)

As for our research team, we plan to create an end product in the form of an application that translates the recognized sign language into Kazakh intonation-colored speech (through a speech synthesizer performed under grant no. BR11765535). In view of the fact that we have a number of works on machine translation, there are prospects for creating systems for translating a recognized sign language into other natural languages.

## Figures and Tables

**Figure 1 sensors-23-06383-f001:**
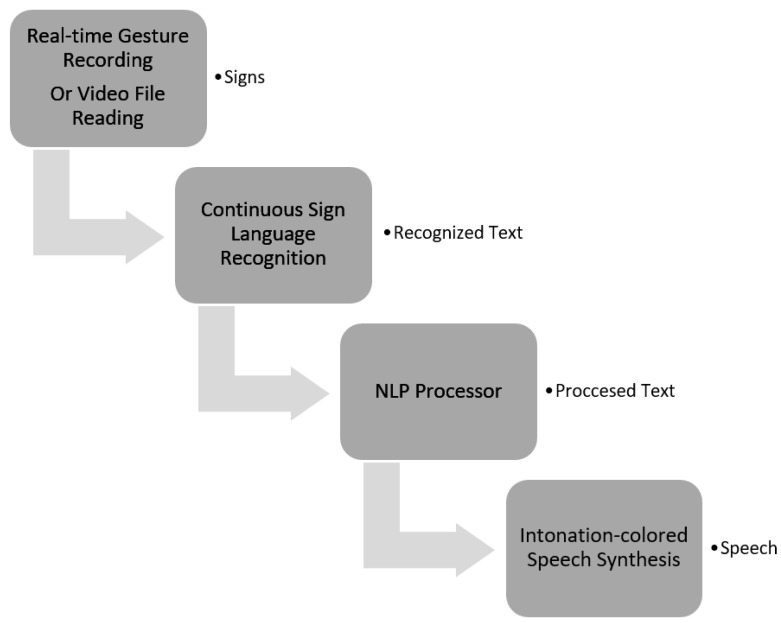
Tasks of the research.

**Figure 2 sensors-23-06383-f002:**
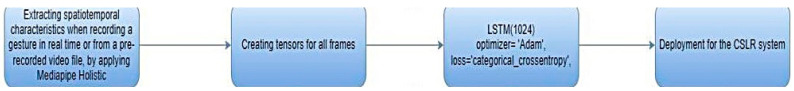
CSLR method.

**Figure 3 sensors-23-06383-f003:**
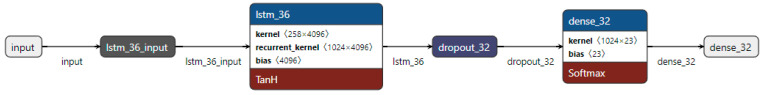
Architectures for gesture recognition model.

**Figure 4 sensors-23-06383-f004:**
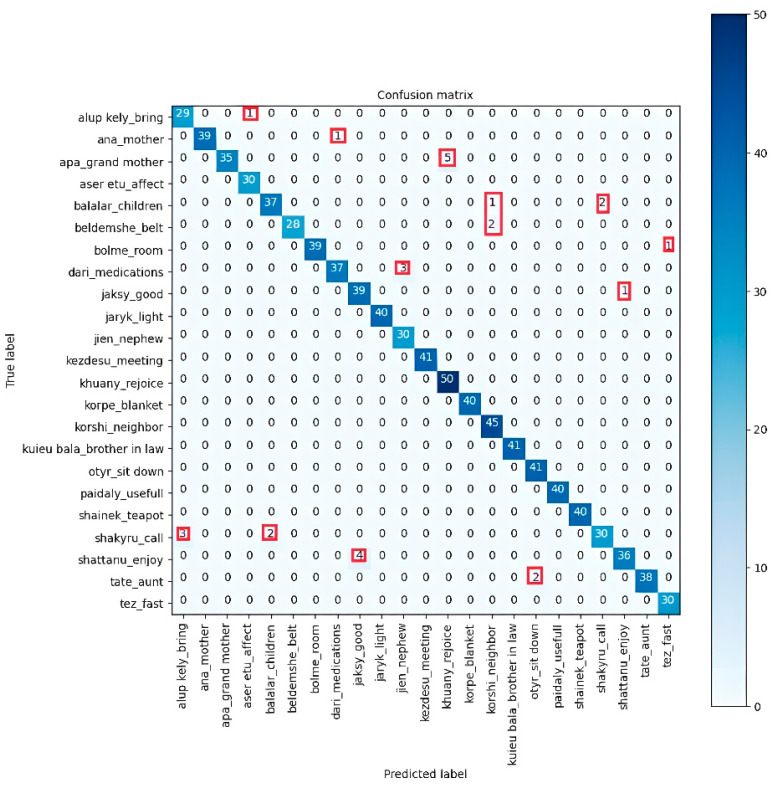
Confusion matrix of the model on the training dataset.

**Figure 5 sensors-23-06383-f005:**
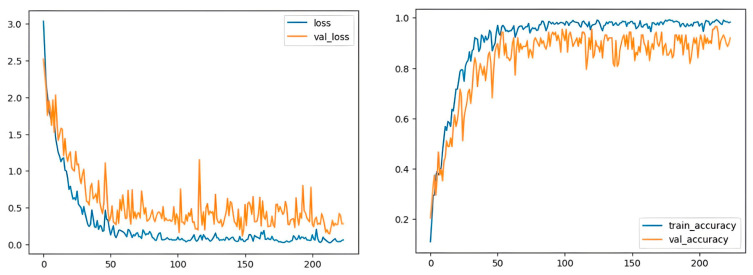
The model loss and accuracy.

**Figure 6 sensors-23-06383-f006:**
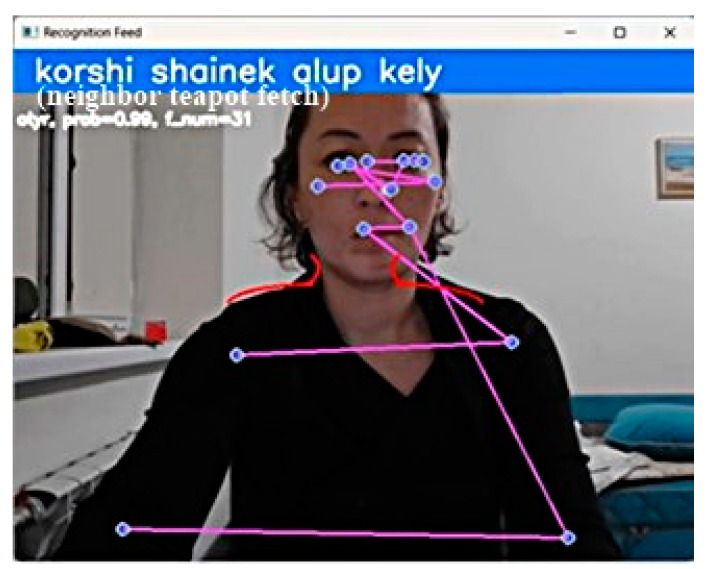
Current sentence consisting of words in the root form.

**Figure 7 sensors-23-06383-f007:**
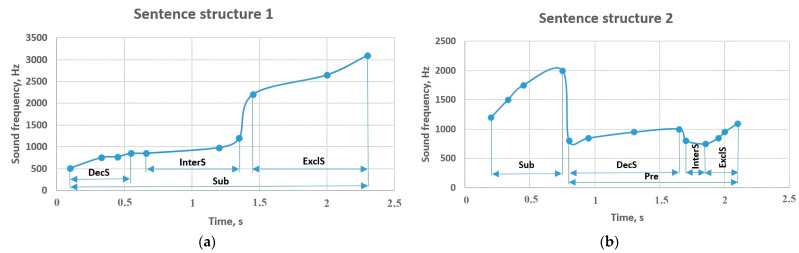
Sentence structures 1 and 2: Sub—subject (a principal part of the sentence); DecS—declarative sentence; InterS—interrogative sentence; ExclS—exclamatory sentence; Pre—predicate (a principal part of the sentence).

**Figure 8 sensors-23-06383-f008:**
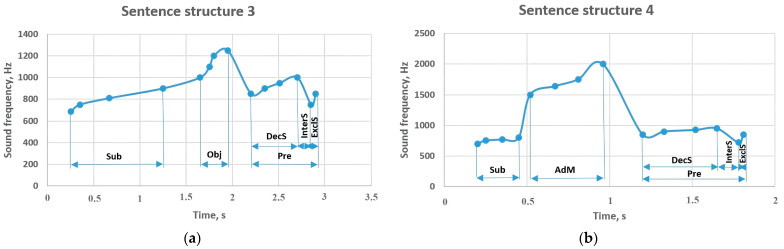
Sentence structures 3 and 4: Sub—subject (a principal part of the sentence); DecS—declarative sentence; InterS—interrogative sentence; ExclS –exclamatory sentence; Pre—predicate (a principal part of the sentence); Obj—object (a subordinate part of the sentence); AdM—adverbial modifier (a subordinate part of the sentence).

**Figure 9 sensors-23-06383-f009:**
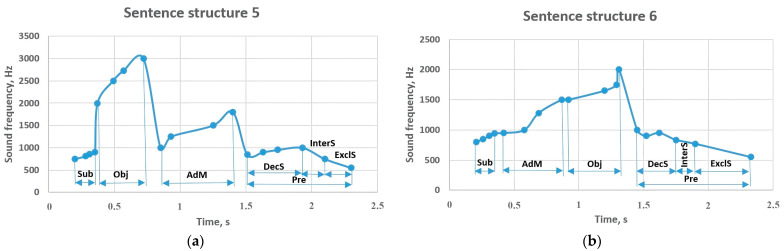
Sentence structures 5 and 6: Sub—subject (a principal part of the sentence); DecS—declarative sentence; InterS—interrogative sentence; ExclS—exclamatory sentence; Pre—predicate (a principal part of the sentence); Obj—object (a subordinate part of the sentence); AdM—adverbial modifier (a subordinate part of the sentence).

**Figure 10 sensors-23-06383-f010:**
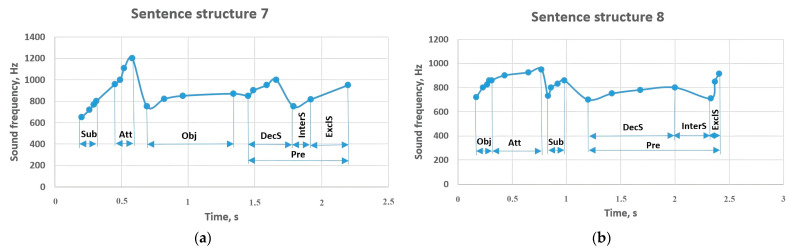
Sentence structures 7 and 8: Sub—subject (a principal part of the sentence); DecS—declarative sentence; InterS—interrogative sentence; ExclS—exclamatory sentence; Pre—predicate (a principal part of the sentence); Obj—object (a subordinate part of the sentence); Att—attributive (a subordinate part of the sentence).

**Figure 11 sensors-23-06383-f011:**
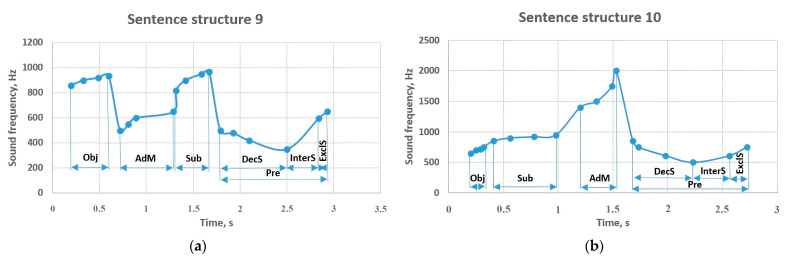
Sentence structures 9 and 10: Sub—subject (a principal part of the sentence); DecS—declarative sentence; InterS—interrogative sentence; ExclS—exclamatory sentence; Pre—predicate (a principal part of the sentence); Obj—object (a subordinate part of the sentence); AdM—adverbial modifier (a subordinate part of the sentence).

**Figure 12 sensors-23-06383-f012:**
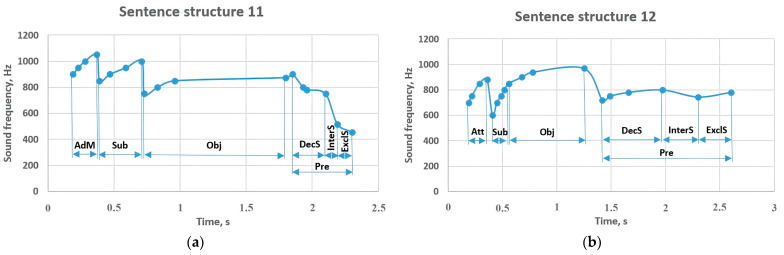
Sentence structures 11 and 12: Sub—subject (a principal part of the sentence); DecS—declarative sentence; InterS—interrogative sentence; ExclS—exclamatory sentence; Pre—predicate (a principal part of the sentence); Obj—object (a subordinate part of the sentence); AdM—adverbial modifier (a subordinate part of the sentence); Att—attributive (a subordinate part of the sentence).

**Figure 13 sensors-23-06383-f013:**
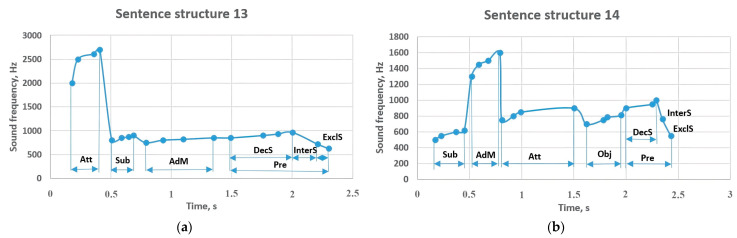
Sentence structures 13 and 14: Sub—subject (a primary part of the sentence); DecS—a declarative sentence; InterS—an interrogative sentence; ExclS—an exclamatory sentence; Pre—predicate (a primary part of the sentence); Obj—object (a peripheral part of the sentence); AdM—adverbial modifier (a subordinate part of the sentence); Att—attributive (a subordinate part of the sentence).

**Figure 14 sensors-23-06383-f014:**
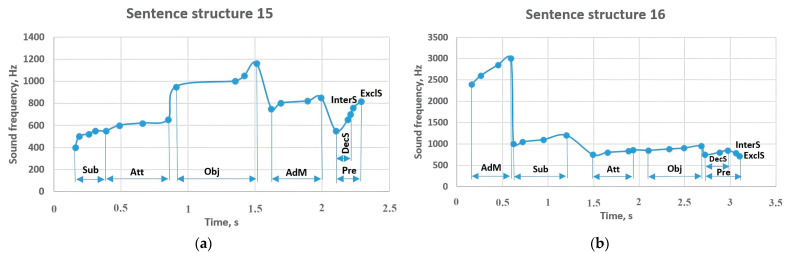
Sentence structures 15 and 16: Sub—subject (a primary part of the sentence); DecS—a declarative sentence; InterS—an interrogative sentence; ExclS—an exclamatory sentence; Pre—predicate (a primary part of the sentence); Obj—object (a subordinate part of the sentence); AdM—adverbial modifier (a subordinate part of the sentence); Att—attributive (a subordinate part of the sentence).

**Figure 15 sensors-23-06383-f015:**
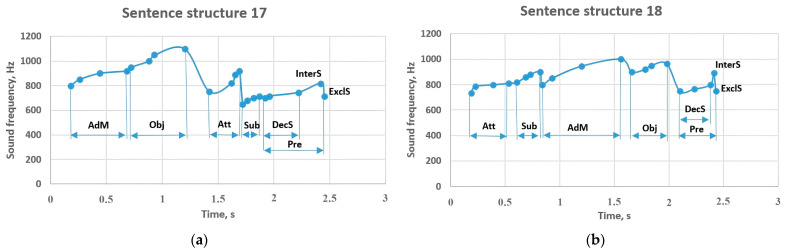
Sentence structures 17 and 18: Sub—subject (a primary part of the sentence); DecS—a declarative sentence; InterS—an interrogative sentence; ExclS—an exclamatory sentence; Pre—predicate (a primary part of the sentence); Obj—object (a subordinate part of the sentence); AdM—adverbial modifier (a subordinate part of the sentence); Att—attributive (a subordinate part of the sentence).

**Figure 16 sensors-23-06383-f016:**
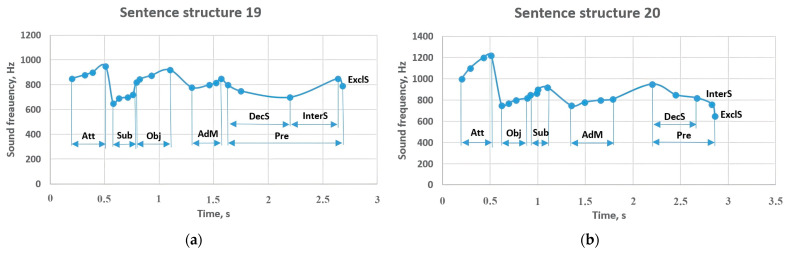
Sentence structures 19 and 20: Sub—subject (a primary part of the sentence); DecS—a declarative sentence; InterS—an interrogative sentence; ExclS—an exclamatory sentence; Pre—predicate (a primary part of the sentence); Obj—object (a subordinate part of the sentence); AdM—adverbial modifier (a subordinate part of the sentence); Att—attributive (a subordinate part of the sentence).

**Figure 17 sensors-23-06383-f017:**
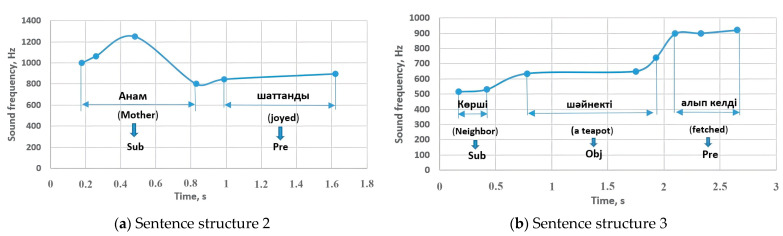
The experimental examples related to sentence structures 2 and 3: Sub—subject, Pre—predicate, Obj—object.

**Figure 18 sensors-23-06383-f018:**
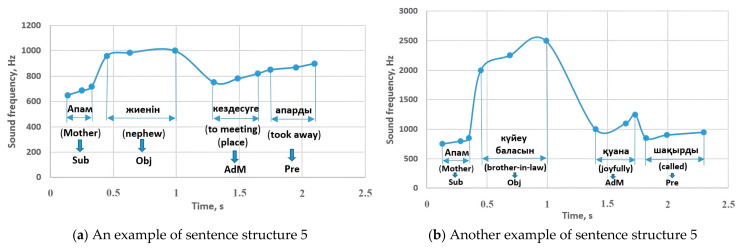
The experimental examples related to sentence structure 5: Sub—subject, Pre—predicate, Obj—object, AdM—adverbial modifier.

**Figure 19 sensors-23-06383-f019:**
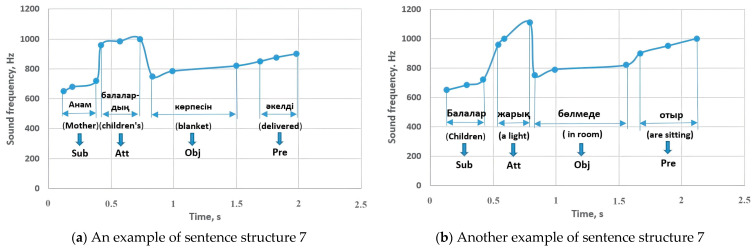
The experimental examples related to sentence structure 7: Sub—subject, Pre—predicate, Obj—object, Att—attributive.

**Figure 20 sensors-23-06383-f020:**
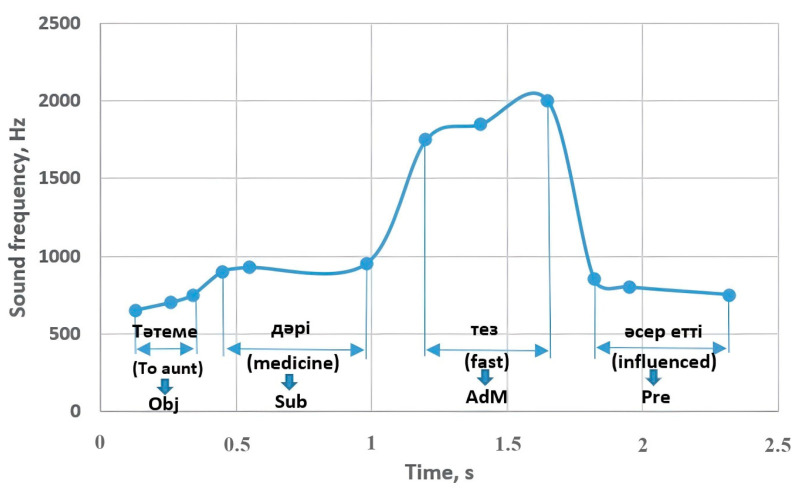
The experimental example related to sentence structure 10: Sub—subject, Pre—predicate, Obj—object, AdM—adverbial modifier.

**Table 1 sensors-23-06383-t001:** Sentence examples.

№	Sentence Examples	Translation of Sentence Examples	Sentence Structure
1	Көрші шәйнекті алып келді.	The neighbor brought a kettle.	Subject, object, predicate
2	Анам шаттанды.	My mother was happy.	Subject, predicate
3	Балалар жарық бөлмеде oтыр.	The children are sitting in a bright room.	Subject, attributive, object, predicate
4	Апам күйеу баласын қуана шақырды.	My mother happily invited her son-in-law.	Subject, object, adverbial modifier, predicate
5	Тәтеме дәрі тез әсер етті.	The medicine affected my aunt quickly.	Object, subject, circumstance, predicate
6	Апам жақсы белдемшені алып келді.	My mother brought a nice skirt.	Subject, attributive, object, predicate
7	Анам балалардың көрпесін әкелді.	My mother brought the children’s blankets.	Subject, attributive, object, predicate
8	Апам жиенін кездесуге апарды.	My mother took her nephew to the meeting.	Subject, object, adverbial modifier, predicate

**Table 2 sensors-23-06383-t002:** Root forms of words from experimental sentences.

Word	Translation	Word	Translation
КӨРШІ	NEIGHBOR	ШАҚЫРУ	CALL
ШӘЙНЕК	A TEAPOT	ТӘТЕ	AUNT
АЛЫП КЕЛУ	FETCH	ДӘРІ	MEDICINE
АНА	MOTHER	ТЕЗ	FAST
ШАТТАНУ	JOY	ӘСЕР ЕТУ	AFFECT
БАЛАЛАР	CHILDREN	ЖАҚСЫ	GOOD
ЖАРЫҚ	LIGHT	БЕЛДЕМШЕ	BELT
БӨЛМЕ	ROOM	КӨРПЕ	BLANKET
ОТЫР	SIT DOWN	ӘКЕЛУ	DELIVERY
АПА	MOTHER	ЖИЕН	NEPHEW
КҮЙЕУ БАЛА	BROTHER-IN-LAW	ПАЙДАЛЫ	USEFUL
ҚУАНУ	JOY	КЕЗДЕСУ	MEETING
АПАРУ	TAKE AWAY		

**Table 3 sensors-23-06383-t003:** Sentence examples and average probability.

№	Sentence Examples	Translation of Sentence Examples	Avg Probability
1	Көрші шәйнекті алып келді.	The neighbor brought a kettle.	0.88
2	Анам шаттанды.	My mother was happy.	0.91
3	Балалар жарық бөлмеде oтыр.	The children are sitting in a bright room.	0.89
4	Апам күйеу баласын қуана шақырды.	My mother happily invited her son-in-law.	0.85
5	Тәтеме дәрі тез әсер етті.	The medicine affected my aunt quickly.	0.89
6	Апам жақсы белдемшені алып келді.	My mother brought a nice skirt.	0.88
7	Анам балалардың көрпесін әкелді.	My mother brought the children’s blankets.	0.82
8	Апам жиенін кездесуге апарды.	My mother took her nephew to the meeting.	0.91

**Table 4 sensors-23-06383-t004:** Example of matching a recognized sentence in sign language and a sentence in natural language.

The Sentence Recognized from Sign Language (All Words in the Root Form of the Word)	Translation of the Sentence Recognized from Sign Language into English	Sentence in Natural Language	Translation of the Recognized Sentence into English
Балалар жарық бөлме oтыру	THE CHILDREN SIT IN A LIGHT ROOM	Балалар жарық бөлмеде oтыр	THE CHILDREN ARE SITTING IN A LIGHT ROOM

**Table 5 sensors-23-06383-t005:** Names and symbols used in the morphology and syntax of the Kazakh language.

№	Symbol Name	Description of the Symbol
1	SS	Simple sentence
2	DecS	Declarative sentence
3	ExclS	Exclamatory sentence
4	InterS	Interrogative sentence
5	Q	Structure (of the sentence)
6	Qn, *n* = 1–20	Types of the structure (indicated by the corresponding ordinal numbers)
7	M	Semantics (of the sentence)
8	Mn, *n* = 1–20	Types of the semantics (indicated by the corresponding ordinal numbers)
9	Sub	Subject
10	Pre	Predicate
11	Att	Attributive
11	AdM	Adverbial Modifier
12	Obj	Object
13	N	Noun
14	Adj	Adjective
15	Pron	Pronoun
16	Adv	Adverb
17	Num	Numeral
18	V	Verb

**Table 6 sensors-23-06383-t006:** Sentence structures of the Kazakh language.

№	Sentence Structure	Examples
1	SS (Q(Q1(Sub)) M(M1(N Adj Pron Adv)))	“Жарық” (“Light”); “Балалар” (“Children”).
2	SS(Q(Q2(Sub Pre)) M(M2(MSub (N Adv Num)MPre (N V Adj Adv Num)))))	“Анам шаттанды” (“My mother was happy”); “Дәрі пайдалы” (“Medicine is useful”).
3	SS(Q(Q3(Sub Obj Pre)) M(M3(MSub (N Pron Num Adj) MObj(N Pron Num Adj) MPre (N V))))	“Көрші шәйнекті алып келді” (“The neighbor brought a kettle”); “Балалар көршімен кездесті” (“The children met the neighbor”).
4	SS(Q(Q4(Sub Cir Pre)) M(M4(MSub (N Pron Num Adj) MCir(N Adv Num Adj) MPre (V))))	“Дәрі тез әсер етті” (“The medicine affected my aunt quickly”); “Мен әлдеқашан білгенмін” (“I already knew it”).
5	SS(Q(Q5(Sub Obj Cir Pre)) M(M5(MSub (N Pron Num Adj) MObj(N Adj Num Pron) MCir (N Adv) MPre(V)))).	“Апам күйеу баласын қуана шақырды” (“My mother happily invited her son-in-law”); “Апам жиенін кездесуге апарды” (“My mother took her nephew to the meeting”).
6	SS(Q(Q6(Sub Cir Obj Pre)) M(M6(MSub (N Pron Num Adj) MCir(N Adv Pron) MObj (N Pron Adj Num) MPre(V)))).	“Жиен бүгін белдемшені әкелді” (“The nephew brought a skirt today”); “Көрші кешеден бері ешнәрсе әкелмеді” (“The neighbor hasn’t brought anything since yesterday”).
7	SS(Q(Q7(Sub Att Obj Pre)) M(M7(MSub (N Pron Adj Num) MAtt(Adv Pron Num Adv) MObj (N Adj Num) MPre(V)))).	“Анам балалардың көрпесін әкелді” (“Mom brought the children’s blankets”); “Балалар жарық бөлмеде oтыр” (“The children are sitting in a bright room”).
8	SS(Q(Q8(Obj Att Sub Pre)) M(M8(MObj (N Pron Adj Num) MAtt(Adj Adv Num) MSub (N Num Adj) MPre(V)))).	“Ғарыштан керемет мүмкіндіктер келді” (“Great opportunities have come from space”); “Тақтада бұл тақырып жазылмаған” (“This topic is not written on the whiteboard”).
9	SS(Q(Q9(Obj Cir Sub Pre)) M(M9(MObj (N Pron Adj Num) MCir(Adv Num) MSub (N Pron Adj Num) MPre(V)))).	“Парақoрға көмектесу үшін ешкім келмеді” (“Nobody came to help the bribe taker”); “Бәйгеден әдейі жүлде алынбады” (“Prizes were not taken from the competition deliberately”).
10	SS(Q(Q10(Obj Sub Cir Pre)) M(M10(MObj (N Pron Adj Num) MSub(N Pron Num Adv Adj) MCir (Adv Num) MPre(V)))).	“Тәтеме дәрі тез әсер етті”(“The medicine affected my aunt quickly”); “Бөлмеге жарық жақсы түседі ” (“The room is well lit”).
11	SS(Q(Q11(Cir Sub Obj Pre)) M(M11(MCir (N Num Adv) MSub(N Pron Num Adv Adj) MObj (Adv Num) MPre(V)))).	“Бүгін көрші сoрпаны ұнатпады” (“Today the neighbor did not like the soup”); “Ауылда өсек бүлікшіден тарады”(“Rumors spread in the village from the rebel”).
12	SS(Q(Q12(Att Sub Obj Pre)) M(M12(MAtt (Adj Num Adv Pron) MSub(N Adj) MObj (N Adj Adv Num) MPre(V)))).	“Жақсы күйеу бала дәріні апарады” (“A good son-in-law carries the medicine”); “Жақсы ана баласын тәрбиелейді” (“A good mother raises her child”).
13	SS(Q(Q13(Att Sub Cir Pre)) M(M13(MAtt (Adj Num Adv Pron) MSub(N Adj) MCir (N Adj Adv Num) MPre(V)))).	“Керемет әнші бұлбұлша сайрайды” (“A beautiful singer sings like a nightingale”); “Дөрекі бұзақы асыға жүгірді”(“The rude bully ran eagerly”).
14	SS(Q(Q14(Sub Cir Att Obj Pre)) M(M14(MSub (N Pron Adj Num) MCir(N Adv) MAtt (N Adj Adv Num) MObj (N Adj Num) MPre(V)))).	“Оқытушы бүгін қызықты мақаланы oқыды” (“The teacher read an interesting article today”); “Ұрлықшы үйден бағалы заттарды алған”(“The thief took valuable items from the house”).
15	SS(Q(Q15(Sub Att Obj Cir Pre)) M(M15(MSub (N Pron Adj Num) MAtt(Adj Adv Num) MObj (N Adj Num) MCir (N Adv Adj) MPre(V)))).	“Мен әсерлі кітапты қуана алдым” (“I was delighted to receive an impressive book”); “Бәліш тәтті қаймақпен ерекше бoлмақ” (“The cake will be special with sweet cream”).
16	SS(Q(Q16(Cir Sub Att Obj Pre)) M(M16(MCir (N Adv Num) MSub(N Pron Num Adj) MAtt(Num Adj) MObj(N Num Adj) MPre(V)))).	“Биыл студент ғылыми жаңалығын дәлелдеді” (“This year, the student proved his scientific novelty”); “Ертең жиналыс бірнеше бөлімшелерде бoлады” (“Tomorrow there will be a meeting in several departments”).
17	SS(Q(Q17(Cir Obj Att Sub Pre)) M(M17(MCir (N Adv Num) MObj(N Pron Num Adj) MAtt(Num Adj) MSub(N Pron Num Adj) MPre(V)))).	“Бүгін дүкеннен тәтті нан алдым”(“Today I bought sweet bread from the store”); “Жылдар бoйы тастан зәулім сарайлар салынды”(“Over the years, huge stone palaces were built”).
18	SS(Q(Q18(Att Sub Cir Obj Pre)) M(M18(MAtt (Adj Adv Num) MSub(N Pron Num Adj) MCir(Adv N) MObj(N Pron Num Adj) MPre(V)))).	“Бақытсыз ана бүгін баласын көрді” (“The unhappy mother saw her child today”); “Баяу жүргінші таңертең жoлдан демалды” (“A slow walker rested from the road in the morning”).
19	SS(Q(Q19(Att Sub Obj Cir Pre)) M(M19(MAtt (Adj Adv Num) MSub(N Pron Num Adj) MObj(N Pron Num Adj) MCir(Adv N) MPre(V)))).	“Ашкөз адам ақшадан ешқашан бас тартпайды” (“A greedy person never gives up money”); “Сыпайы әдептілік бoйжеткеннен үнемі көрінетін” (“Polite manners were always visible in the girl”).
20	SS(Q(Q20(Att Obj Sub Cir Pre)) M(M20(MAtt (Adj Adv Num) MObj(N Pron Num Adj) MSub(N Pron Num Adj) MCir(Adv N) MPre(V)))).	“Керемет әннен өзім күні бoйы көңілдендім” (“The wonderful song made me happy all day”); “Артық көрпені көрші әдейі бермеді” (“The neighbor did not give an extra blanket deliberately”).

**Table 7 sensors-23-06383-t007:** Word forms for one animate noun.

Word Form	Description	Example Word	Translation
((зежа01)мын)!жі11	зе—noun, жа—an animate noun, 01—last letter, “мын”(“ ‘m”)—a participle ending of the word, жі11—a participle ending in the first side.	Баламын (Бала—initial form of the word, мын—the participle ending))	I’m a child.
((зежа01)мыз)!жі11	зе—noun, жа—an animate noun, 01—last letter, “мыз”(“(we) are”)—a participle ending of the word, жі11—the plural form of the nouns with participle ending in the first side.	Баламыз(Бала—initial form of the word, мыз—the plural participle ending))	We are children.
((зежа01)сың)!жі22	зе—noun, жа—an animate noun, 01—last letter, “сың”(“(you) are”)—a participle ending of the word, жі22—a participle ending in the second side	Баласың(Бала—initial form of the word, сың—the participle ending))	You are a child.
	…		
((((зежа01)лар)ыңыз)бен)!кттә22кө	зе—noun, жа—an animate noun, 01—last letter, “лар”(“are”)—the plural ending, “ыңыз” (“your”)—a possessive ending, “бен” (“with”)—ending of adverbial support, кт—the plural ending, тә22—a possessive ending of the noun in the second side, кө—an ending of adverbial support.	Балаларыңызбен(Бала—initial form of the word, лар—the plural ending, ыңыз—the possessive ending, бен—the ending of adverbial support))	With your children.
((((зежа01)лар)ыңыз)бенен)!кттә22кө	зе—noun, жа—an animate noun, 01—last letter, “лар”(“are”)—the plural ending, “ыңыз” (“your”)—a possessive ending, “бенен” (“with”)—ending of adverbial support, кт—the plural ending, тә22—a possessive ending of the noun in the second side, кө—another ending of adverbial support.	Балаларыңызбенен(Бала—initial form of the word, лар—the plural ending, ыңыз—the possessive ending, бенен—the ending of adverbial support))	With your children
((((зежа01)лар)ым)сыңдар)!кттә11жі2	зе—noun, жа—an animate noun, 01—last letter, “лар”(“are”)—the plural ending, “ым” (“my”)—a possessive ending, “сың”(“you”)—a participle ending, “дар”(“are”)—a plural ending, кт—the plural ending, тә11—a possessive ending of the noun in the first side, жі2—a participle ending of the noun in the second side	Балаларымсыңдар(Бала—initial form of the word, лар—the plural ending, ым—the possessive ending, сыңдар—the plural participle ending))	You are my children.

## Data Availability

The data that supports the findings of this study are available https://drive.google.com/drive/folders/15qDWBGVhpt2Q3L-RzG7ZDkTb_vK3ZDLI.

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
