# Peer review of "Continuous Sign Language Recognition and Its Translation into Intonation-Colored Speech"

_sensors, 2023, doi:10.3390/s23146383_

Round 1

Reviewer 1 Report

The overall research question is interesting. These are some of the questions/ideas I had while reading the paper. Perhaps emphasizing these points will help improve the readability of the paper/quality of the work. 

- The labels in Figure 4 are too blurry to read. Please improve.

- What is the vocabulary size of the CSLR? Is it the words in Table 2? If that's the case, the vocabulary seems too small to be able to communicate effectively. Or is the vocabulary sub-word where the sign language user has to spell each word?

- I am not so sure information what Figures 7-20 are supposed to convey. What is the unit of the x-axis? The graphs are neither time-domain nor frequency-domain plot. Nor do they look like spectrum/spectrogram. Since the purpose of the system is to generate speech from sign language, maybe it would be better to compare the spectrogram (such as Short Time Fourier Transform) of the sound generated by the system compared to the speech of a normal speaker saying the same sentence?

- To demonstrate the effectiveness of intonation speech synthesis, is it possible to pass the resulting speech through a speech recognition model and see if it successfully recognizes the intended message? The word error rate of the speech recognition model can then be used to measure the overall performance of the system. I recognize that this is a lot of work so it's just a suggestion.

Author Response

Thank you for reviewing our research paper and for your comments, which helped us to get our writing correct and improve it. We have taken all the comments into account and made corrections according to them.

Reviewer 2 Report

The authors presented a methodology for recognizing sign language and posterior conversion into intonated speech signals. The paper proposes an interesting application that will (potentially) improve people's communication. I'm leaning towards accepting the paper; however, it can benefit from some revision before considering it for publication.

Major concerns:

  • The introduction should start with the definition of the problem, how it has been addressed, and finally, what the proposed approach is. The current version of the introduction begins with a summary of the methodology followed by the authors, which I think should be in the first part of the section "Methods."
  • I suggest explicitly writing this work's contributions in a subsection after the related work. Also, make the related work a subsection of the introduction.
  • The speech synthesis approach needs a better metric to measure precision. Commonly, people use the mean opinion score (MOS) for evaluation.
  • I'm not convinced about using the "Sound frequency" to analyze intonation and emotion. If I understood correctly, this is simply the sum of the energy components of speech segments; however, the phonetic content can vary depending on the sounds analyzed; thus, the graph seems more related to the sounds than the prosody. Please clarify how this measure works and how we can be sure that is the emotion/intonation we see in Figures 7 to 20.
  • There should be a "Data" section describing the type of recordings used, how they were acquired, the number of participants, and information about the setup used for the acquisition. At the moment, the data description is all over the manuscript.

Minor comments:

  • The abstract should contain information about the data considered for the study, the number of subjects, and the main results for sign language recognition, the NLP processor, and speech synthesis.
  • Some abbreviations are not defined, e.g., SL, CSLR, NLP.
  • Figure 4 is too small. It is not possible to read the labels or the numbers.
  • Figures 18.a and 19.a are the same, even though these are supposed to display different information. Is this an error? 
  • Page 15. "Morphological corrector (word modifier)" should be formatted as a subsection section (3.5).
  • Page 6. The sentence "The obtained features were trained using a single LSTM layer.." is unclear. Was the LSTM trained with the gesture data? Please rephrase.
  • There should be a short description of the model/tool used to automatically detect the region of interest (ROI).

The English is clear enough. 

Author Response

(The authors gave the same response as above.)

Reviewer 3 Report

The authors propose a method for Sign Language Recognition that also translates the intonation of the speech.

The idea is very very interesting.

The paper needs some improvements

Abstract. Put here also quantitative results and main novelties.

Introduction. Creates different paragraphs/subsections where to highlight motivations and objectives, research questions, deficiency of literature, gaps filled by your work, proposed approach in brief, main novelties and results, major contributions, structure of the paper. Figure 1 should not be in the introduction. I suggest to put there a visual abstract of the proposal, not the methodology or its tasks.

Background. Please add a background section where to provide all the basics to understand the methods applied in your paper.

Related works. I suggest the authors to improve this section. For example, it could be split into subsections based on the tasks in Figure 1. However, authors should provide a point-to-point comparison with each of the related works and highlight those that used publicly available datasets or released one. Indeed, (this can be considered a point related to the experimental part of the paper) I expect in the experimental part direct performance comparison of the proposal on different datasets and indirect comparison with related works. Furthermore, in related work section, a table summarizing weaknesses and strenghts, pros and cons of previously proposed approach compared to yours should be put at the end of the section. Lastly, in related work authors should account for some of the latest papers on gesture recognition (a brief overview) in different contexts, from soft biometrics (gender, age) to techno-regulation (age), from human-robot interaction to handwriting.

Dataset. It should be made available entirely or at least some excerpt.

Gesture recognition model. Motivate the choice of such an architecture. What kind of experiments have been conducted to reach that architecture?

Figures. (must) Put them in high resolution. Figure 4 must be larger (1.5 times or twice)

Validation. "[...] To obtain an effective and gender-resistant model, each word from the sentence was recorded by 5 demonstrators (a 9-year-old girl, a 12-year-old boy, a 40-year-old woman and two 20-year-old girls) 50 samples for a training sample, 10 samples for a test sample and 5 samples for a validation sample. [...]" Why do not apply Leave-One-Out or Leave-One-USER-Out cross-validation in this paper (see https://doi.org/10.1016/j.eswa.2023.119614)? Motivate.

Figure 4. Discuss errors highlighted.

Pages 12-14. Please find another way to insert those details. I suggest to make them more visually enhanced. It's hard to understand what is there.

Discussion. Please improve this section

Limitations. please add this section.

Expected impact. Add this section (economic, social, technological impacts must be discussed)

Future works. Please add this section.

Please proofread the paper before the revised version submission. There are some typos and grammar errors.

Author Response

(The authors gave the same response as above.)

Round 2

Reviewer 2 Report

I thank the authors for addressing my comments. However, there are some aspects that still need improvement.

1. The resolution of the figures is rather low. The text is blurry in most of them.

2. Regarding my comment on the datasets, the authors responded: "According to the authors, in view of the fact that the study is complex and consists of several stages with different datasets, the inclusion of a dataset section in each method makes the article unreadable. Highlighting the general section of the dataset outside of the calculations also reduces the perception of the article by readers. Therefore, the description of datasets is included in the text of the article where necessary."

I refuse to believe that this would take too much time, especially if the data is already described in the manuscript. It shouldn't take more than one page to clearly describe how the data was captured, labeled, and how many subjects were considered.  The current manuscript makes it confusing.

3. Regarding the synthesis, I mentioned it because that is how you refer to it in the paper. So I suggest not using the word synthesis to refer to intonation measurements. That's very confusing.

4. The abbreviations are now defined, but you need to reference them the first time these are mentioned. For instance natural language processing (NLP).

Author Response

(The authors gave the same response as above.)

Reviewer 3 Report

1. There is still no visual abstract for the paper.

2. My comment: Introduction. Creates different paragraphs/subsections where to highlight motivations and objectives, research questions, deficiency of literature, gaps filled by your work, proposed approach in brief, main novelties and results, major contributions, structure of the paper. Figure 1 should not be in the introduction. I suggest to put there a visual abstract of the proposal, not the methodology or its tasks.

Authors response: We have taken your comment into account, thank you.

My new comment: authors should still add those information in the introduction of the paper. Moreover, the introduction is not backed by proper citations/literature.

3. My comment: Gesture recognition model. Motivate the choice of such an architecture. What kind of experiments have been conducted to reach that architecture?

Authors response: As part of the study, the 2DCNN and LSTM1024 architectures were chosen. When integrated into a real continuous recognition system, unlike the 2DCNN model, the LSTM1024 model showed a better gesture prediction result, because the LSTM can work with sequential or temporal data and has the ability to retain and use information about past events to make decisions in the present, which is important for continuous real-time gesture recognition.

My new comment: this response is not satisfactory. Why 2DCNN and LSTM1024 were chosen? Please, justify the choices (also through experiments).

Please, proofread the paper.

Author Response

(The authors gave the same response as above.)
